# Pairwise library screen systematically interrogates *Staphylococcus aureus* Cas9 specificity in human cells

Josh Tycko [1,5], Luis A. Barrera[1,6], Nicholas C. Huston[1,7], Ari E. Friedland[1], Xuebing Wu[2], Jonathan S. Gootenberg [3], Omar O. Abudayyeh[4], Vic E. Myer[1], Christopher J. Wilson [1] & Patrick D. Hsu[1,8]

Therapeutic genome editing with *Staphylococcus aureus* Cas9 (SaCas9) requires a rigorous understanding of its potential off-target activity in the human genome. Here we report a high-throughput screening approach to measure SaCas9 genome editing variation in human cells across a large repertoire of 88,692 single guide RNAs (sgRNAs) paired with matched or mismatched target sites in a synthetic cassette. We incorporate randomized barcodes that enable whitelisting of correctly synthesized molecules for further downstream analysis, in order to circumvent the limitation of oligonucleotide synthesis errors. We find SaCas9 sgRNAs with 21-mer or 22-mer spacer sequences are generally more active, although high efficiency 20-mer spacers are markedly less tolerant of mismatches. Using this dataset, we developed an SaCas9 specificity model that performs robustly in ranking off-target sites. The barcoded pairwise library screen enabled high-fidelity recovery of guide-target relationships, providing a scalable framework for the investigation of CRISPR enzyme properties and general nucleic acid interactions.

[1] Editas Medicine, 11 Hurley St., Cambridge, MA 02141, USA. [2] Whitehead Institute for Biomedical Research, Cambridge, MA 02142, USA. [3] Department of Systems Biology, Harvard, Cambridge, MA 02138, USA. [4] Department of Health Sciences and Technology, Massachusetts Institute of Technology, Cambridge, MA 02139, USA. [5] Present address: Department of Genetics, Stanford University School of Medicine, Stanford, CA 94305, USA. [6] Present address: Arrakis Therapeutics, 35 Gatehouse Dr., Waltham, MA 02451, USA. [7] Present address: Department of Molecular Biophysics and Biochemistry, Yale University, New Haven, CT 06511, USA. [8] Present address: Laboratory of Molecular and Cell Biology, Salk Institute for Biological Studies, La Jolla, CA 92037, USA. Correspondence and requests for materials should be addressed to C.J.W. (email: christopher.wilson@editasmed.com) or to P.D.H. (email: patrick@salk.edu)

The compact SaCas9 enables in vivo delivery of Cas9 and multiple single guide RNAs (sgRNA) packaged within a single adeno-associated virus (AAV) vector[1,2], serving as a promising platform for gene editing therapies. AAV-SaCas9 is capable of achieving therapeutic levels of genome editing in preclinical animal models of Duchenne muscular dystrophy[3,4], ornithine transcarbamylase deficiency[5], and of HIV infection[6]. Translating these promising initial results into medicines requires a rigorous understanding of intended and unintended genome editing. SaCas9-mediated off-target effects have been detected with genome-wide methods, including GUIDE-seq[7] and BLESS[1,8], and direct visualization of dSaCas9-EGFP binding in cells[9]. However, the sequence determinants of SaCas9 cleavage specificity have not been profiled. Furthermore, SaCas9 is known to efficiently cleave genomic DNA with spacer lengths from 20 to 24 nt[1,2], but the effect of spacer length on specificity is not known.

To systematically interrogate SaCas9 specificity in human cells, we developed a method to test a library of sgRNAs against a library of genome-integrated synthetic target sequences. Lentiviral delivery of the pairwise library cassette results in integration of a sgRNA and paired synthetic target site in the genome. We designed a library of 88,692 guide-target pairs, distributed among 73 sgRNA groups. Within a group, the sgRNA had shared sequence in positions 1–18 and ranged in length from 19 to 24 nt spacers. All sgRNAs were paired with target sites bearing all possible single mismatches and subsets of sgRNAs were paired with all possible double mismatches or all possible double transversions. Five groups of sgRNA were paired with target sites bearing all possible single insertions and deletions, to study the effect of DNA and RNA bulges. The protospacer adjacent motif (PAM) was held constant at 5′-CAGGGT-3′ to match the consensus sequence of 5′-NNGRRT-3′[1]. This pairwise library design enables high-throughput characterization of SaCas9 in cells, while controlling for the effects of delivery and chromatin context, and allows us to determine the optimal spacer lengths for specific genome editing.

## Results

**Double barcode design improves pairwise screen measurements.** Lentiviral delivery of the pairwise library cassette results in integration of a sgRNA and paired synthetic target site in the genome (Fig. 1a). We designed a library of 88,692 guide-target pairs, distributed among 73 sgRNA groups (Supplementary Table 1). We measured the genome editing activity of these guide-target pairs in human cells (Fig. 1b). The library cassette lentivirus was transduced in HEK 293FT cells at low multiplicity-of-infection (MOI) to enrich for single-copy integration events, ensuring independent editing reactions per cell. Genomic DNA was extracted 0, 3, and 14 days after SaCas9 transduction and the library cassette was PCR-amplified prior to Illumina sequencing.

After editing has occurred, insertions or deletions (indels) in the target site can obscure the original guide-target pair relationship. Accordingly, each library member (i.e., a unique guide-target pair) was linked with a unique, error-correcting Hamming barcode[10] 16 basepairs downstream of the cut site (Supplementary Fig. 1A). Post-editing, this Hamming barcode identifies the original guide-target sequence in a sequence read. The indel frequencies associated with each Hamming barcode determine editing efficiency for each guide-target pair. Across sequencing runs, we found a perfect match to a Hamming code in 83% of reads, on average. After using the error-correcting algorithm to retrieve the missing Hamming codes, we were able to match 23% of the remaining reads to a Hamming code, resulting in ~87% Hamming codes being recovered in total.

Reasoning that a subset of molecules representing a particular library member would be subject to synthesis errors that generate an inappropriate mismatch or indel, we appended an additional randomized barcode (rBC) downstream of the sgRNA. These rBC effectively barcode unique lentiviral integrations. Sequencing from Day 0 was used to whitelist the guide-target cassettes that were error-free and sufficiently represented prior to Cas9 delivery. This whitelist minimizes false positive indels that arise from synthesis errors or other causes, and improves the reproducibility of pairwise library screens by filtering out library members with insufficient cellular representation (i.e., library members with a low number of rBCs).

Indel rates and the corresponding off:on-target ratios at Day 3 and 14 were then computed using only whitelisted rBCs (i.e., cassettes that were error-free at Day 0). Indel levels were observed to be increasingly reproducible as the minimum number of unique whitelist rBCs per library member was also increased (Fig. 1c). After filtering for library members with at least 20 independent integrations (i.e., ≥20 rBCs), 83% of the library remained and the biological replicates were correlated ($R^2 = 0.76$) (Fig. 1d).

**21 and 22 nt spacers are most efficient.** We first analyzed the on-target activity of SaCas9. Based on our observation of saturation in indel levels by Day 6 in a pilot time-course study (Supplementary Fig. 1), we sequenced the screen samples on Day 3 to determine representative but non-saturated indel efficiency, and again on Day 14. 21 and 22 nt spacers edited on-target sites most efficiently at both time points (Fig. 1e and Supplementary Fig. 2A). This result refines smaller-scale studies (of 4 and 21 sgRNAs) that had defined the optimal spacer length as being 21–23 nt[1] or 20–24 nt[2].

**20 nt spacers are less tolerant of mismatches.** Next, we compared the average off-target indel rates for mismatched guide-target library members and found targets with single mismatches were edited 3.4-fold less than matched targets on average ($p <$ 0.01, Dunn's multiple comparisons test) (Fig. 1f), while the targets with double mismatches or bulges were edited even less frequently.

SaCas9 off:on-target activity can be further considered as a function of both mismatch identity and position. Overall, mismatch tolerance was low in the ~9 nt PAM-proximal seed region, oscillated between positions 13–19, and was higher at the PAM-distal region (Fig. 2a, b). This tolerance pattern was consistent across spacer lengths, but the 21 nt spacer was the most tolerant of mismatches at any position. The 20 nt spacer was significantly less tolerant of single mismatches than the 21 or 22 nt spacer at both Day 3 and Day 14 ($p < 0.01$, Dunn's multiple comparisons test, Fig. 2a and Supplementary Fig. 2B).

We considered that the apparent differences in specificity across spacer lengths might be due to the differences in guide efficiency. To study specificity while controlling for only highly active guides, we examined the top ten most active guides within each spacer length group. We did not observe a statistically significant difference between the on-target efficiencies of the top ten guides of the 20, 21, and 22 nt spacer groups (Dunn's multiple comparisons test, Supplementary Fig. 3A), although the 20 nt spacer efficiency did trend lower than the 21 nt spacer efficiency. In this subset, guides with a 20 nt spacer were again less tolerant of single mismatches than the 21 or 22 nt spacer groups at both Day 3 and Day 14 ($p < 0.001$, Dunn's multiple comparisons test, Supplementary Fig. 3B). Notably, these highly active 20 nt spacers are atypical, but this analysis shows that their lower tolerance of mismatches cannot be attributed to low overall activity.

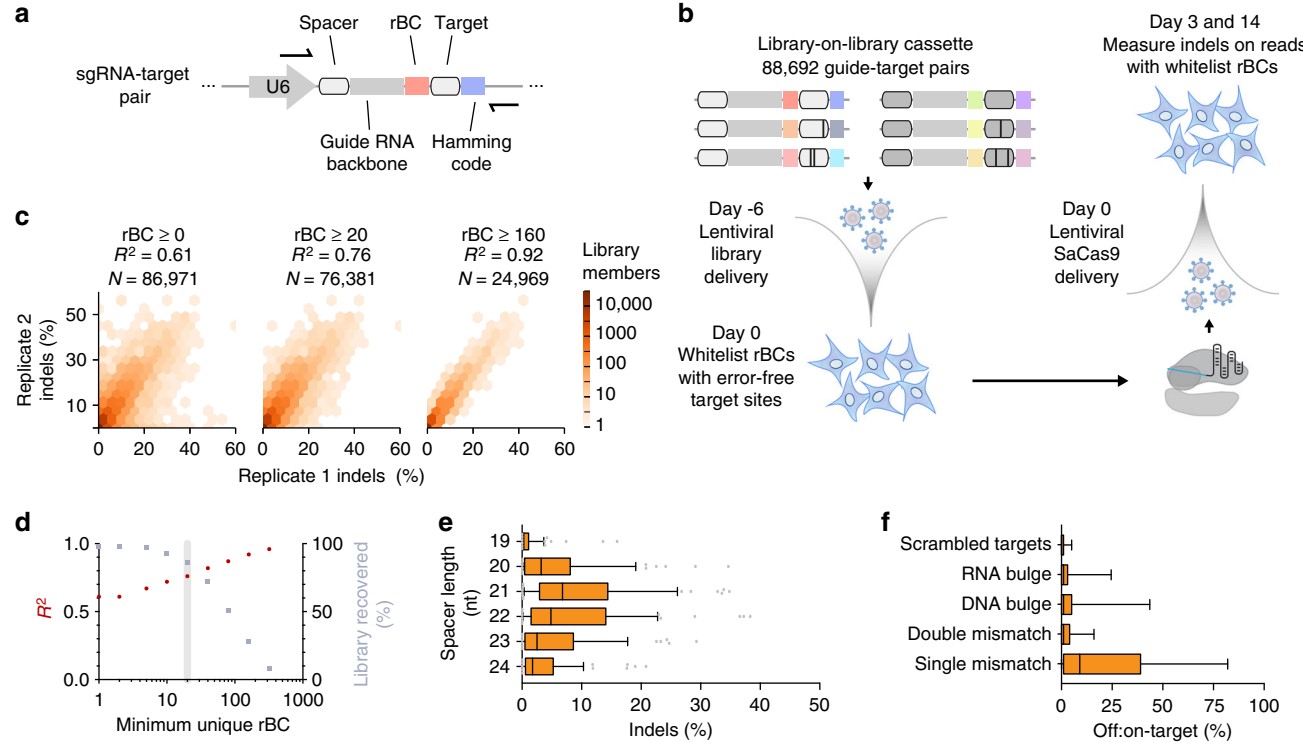

**Fig. 1** Pairwise library screen of SaCas9 genome editing specificity. **a** Schematic of the pairwise library cassette. Individual library members have variable spacer and target sequences, and each member is identifiable by a unique 15 nt error-correcting Hamming barcode. Individual molecules of each library member are tagged with a unique randomized barcode (rBC). **b** Schematic of the pooled pairwise library screen workflow. Each guide-target pair is associated with many rBCs. The library was initially installed in HEK 293T cells by lentivirus and sequenced to generate a whitelist of rBCs associated with error-free guide-target pairs. SaCas9 was then delivered by lentivirus. After 3 and 14 days, the library was sequenced to measure Cas9-mediated indels. **c** Reproducibility of a pairwise library screen increases if a greater number of whitelist rBCs is required for each library member. Heat color represents the number of library members in that hexagonal bin, while white area represents 0 library members. **d** The fraction of recovered library members decreases as a greater number of rBCs is required. All downstream analyses were performed with a minimum of 20 unique whitelist rBCs for each library member (grey). **e** On-target indel efficiency on Day 3 for SaCas9 guide-target pairs, binned by spacer length. (n = 653 guide-target pairs). **f** Comparison of SaCas9 activity across categories of target sites. The scrambled targets are negative controls (n = 47,374 guide-target pairs). **c** through **f** use data from Day 3 post-SaCas9 delivery. Boxes in **e** and **f** denote median and interquartile range (IQR), and whiskers extend to the 10th and 90th percentile

Across the full set of spacers, the 20 nt spacer was also less tolerant of double mismatches than the longer guides ($p < 0.0001$, Dunn's multiple comparisons test, Supplementary Fig. 2C), in a position-dependent manner (Fig. 2c, e). 21 nt spacers had a mean 16% off:on target activity ratio while we observed a mean of only 2% off:on-target activity across all double-mismatched sites targeted by 20 nt spacers (Fig. 2c, d), at Day 3. We observed an increase in double-mismatch tolerance at Day 14, and the 20 nt spacer remained significantly less tolerant than the longer guides ($p < 0.0001$, Dunn's multiple comparisons test, Supplementary Fig. 2C).

Upon inspecting the subset of library members with bulges, (single-nucleotide indels in the target site), we found that both RNA and DNA bulges in the sgRNA:target duplex were minimally tolerated at positions 1–18, regardless of sgRNA length (Supplementary Fig. 4B, C). This bulge intolerance remained consistent on Day 14. However, PAM-distal bulges were near-completely tolerated. The bulge nucleotide identity had no significant effect on indel measurements (Supplementary Fig. 4B, C).

**Validation of spacer length affecting specificity**. Given the effect of the spacer length on mismatch tolerance, we conducted Northern blot analysis to confirm that this range of spacer lengths is reliably maintained in human cells. sgRNA from 18 to 24 nt were accurately maintained at full length when expressed from a U6 promoter, regardless of whether or not the 5′ nucleotide was a guanine (Supplementary Fig. 5). Further, sgRNA were hardly detectable in cells lacking SaCas9 expression.

To validate these findings with an orthogonal method, we designed GFP-targeting sgRNAs with variable length spacers and synthesized sgRNA with single mismatches at every position. In independent wells, we transiently transfected SaCas9 and each sgRNA into a stable HEK 293T-GFP cell line and measured GFP knockout by flow cytometry. Consistent with the screen, we observed low tolerance of PAM-proximal mismatches and more variable PAM-distal tolerance, in a mismatch identity and nucleotide position-dependent manner. As in the screen, we again found 20 nt spacers to be less tolerant of single mismatches (Supplementary Fig. 6).

**SaCas9 specificity score ranks off-target sites**. We next formulated a non-linear regression specificity score, trained from our screen dataset, to integrate the relative contribution of mismatch position, identity, number, and density. These scores correlate well with the observed off-target activity in the screen (Supplementary Fig. 7). Our model assumes that mismatches can be modeled independently and performs similarly on off-targets with single or multiple mismatches (Supplementary Fig. 7). This scoring algorithm was capable of predicting SaCas9 specificity in the orthogonal GFP mismatch assay with high performance (Fig. 3a). Furthermore, the score performed well in ranking

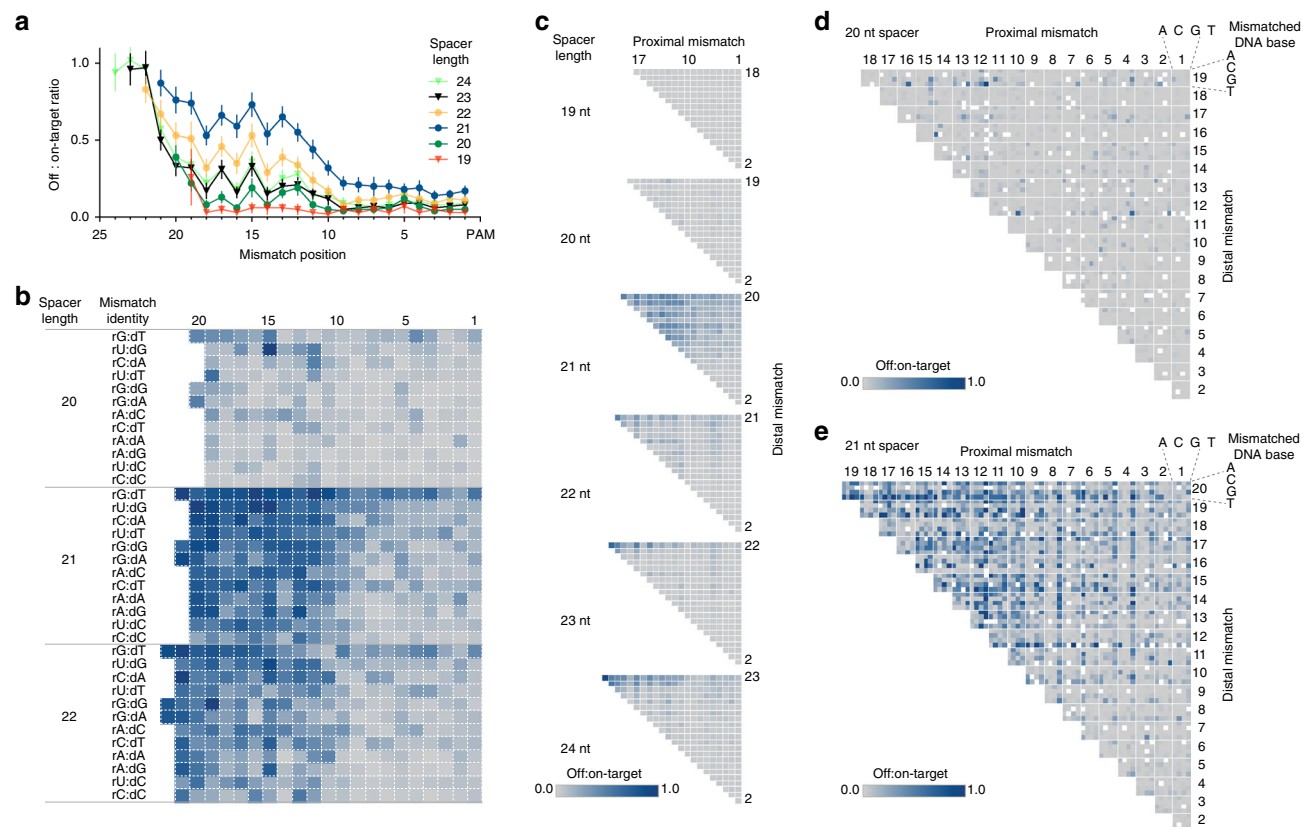

**Fig. 2** SaCas9 sgRNA with shorter spacers are less tolerant of mismatches. **a** Average effect of sgRNA spacer length and mismatch position on SaCas9 single-mismatch tolerance. Mean ± 95% confidence interval is shown (*n* = 16,742 guide-target pairs). **b** Heatmap of relative SaCas9 cleavage efficiency for each possible RNA:DNA base pair, calculated for all single mismatch library members (*n* = 10,344 guide-target pairs). **c** Heatmap of relative SaCas9 double-mismatch tolerance across different spacer lengths and positions. Data is aggregated from 10 sgRNAs for which all possible single and double mismatches were tested (*n* = 28,707 guide-target pairs). **d** DNA base identity and position effect on double-mismatch tolerance for 20 nt spacer sgRNA (*n* = 6017 guide-target pairs) and **e** for 21 nt spacer sgRNA (*n* = 6729 guide-target pairs). All panels use data from Day 3 post-SaCas9 delivery

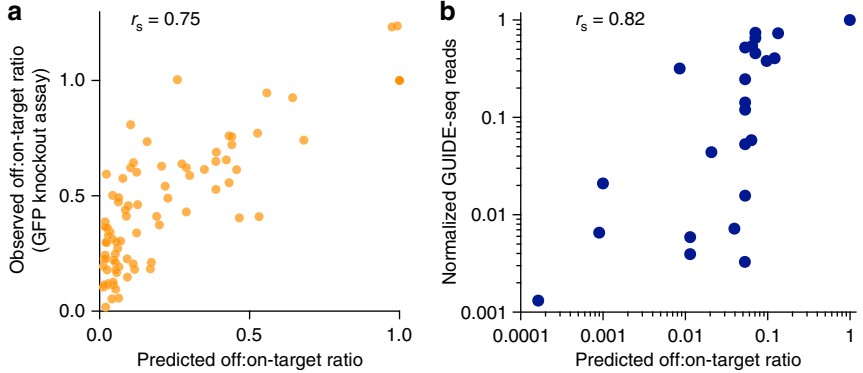

**Fig. 3** SaCas9 specificity model ranks synthetic and endogenous off-target sites. **a** A regression model trained on the pairwise library screen data correlates with an independent validation set (*n* = 86 guide-target pairs). The sgRNAs knock out a stable GFP and include single mismatches in the spacers to mimic off-target effects. **b** Spearman rank correlation of the SaCas9 specificity model compared with previously reported SaCas9 GUIDE-seq data from a different cell type, U2OS[7]

endogenous off-target sites in a different cell type (U2OS), as measured by GUIDE-seq[7,11], with a Spearman correlation of 0.82 (Fig. 3b).

## Discussion

These findings support a simple strategy to mitigate the risk of off-target activity by adjusting the spacer length. 20 nt spacers reduce SaCas9 mismatch tolerance, which parallels the finding that 17–18 nt truncated guide RNA spacers can improve SpCas9 specificity[12–15]. However, these shorter spacers are less efficient on average so more experimental screening may be necessary to find a suitably efficient 20 nt spacer. This strategy can be combined with expanded PAM variants in situations where the number of candidate guides is low[7], such as for targeting of pathogenic SNPs. Our study supports known strategies

such as selecting sgRNAs with maximal sequence dissimilarity from off-target sites and avoiding off-target sites with only PAM-distal mismatches. Importantly, the specificity model trained on SaCas9-specific parameters can be used for in silico selection of guides and to prioritize off-target sites for follow-up.

The challenge of characterizing genome editing and nucleic acid specificity is well-suited to high-throughput approaches because of the large space of possible guide-target pairs. While genome-wide off-target detection methods[1,11,13,16–19] are important for characterizing individual sgRNA of interest, they are also limited by the availability of endogenous off-target sites, and do not provide general models of specificity[20]. A complementary approach is to generate a synthetic library that more thoroughly covers the space of possible off-target sites. To date, such studies have primarily been performed in vitro[21,22] and in yeast[23]. While specificity of the Cas9 from *S. pyogenes* has been previously profiled in human cells via cell surface marker knockout and flow cytometry[24], the pairwise library approach described here provides a more programmable, alternative method. Previous reports suggest that even single-nucleotide changes in a sgRNA can strongly affect activity by changing RNA secondary structure[25], but our screen avoids this concern and mimics a real-world scenario by placing the mismatches in the target DNA. Further, the edits are directly measured at the DNA sequence. This property enables high-throughput evaluation of nuclease-mediated DNA repair outcomes.

A related approach was recently employed to characterize Cpf1 as a genome editing tool[26] by screening linked libraries, but was sometimes limited by high error rates at baseline. In contrast, our design includes two barcodes, one for the target-guide pair and another to track each integration event, which allows us to whitelist the targets that were error-free at baseline and recover the majority of the library for analysis. Low MOI delivery of our pairwise library further facilitates the measurement of independent editing reactions for each guide-target pair compartmentalized in individual cells.

Our barcoded pairwise library screening approach provides a general framework for understanding and engineering nucleic acid interactions, and could be exploited for oligonucleotide probe or switch design. We demonstrate its utility via high-throughput characterization of SaCas9 specificity, which could be extended to interrogate other nuclease properties in future studies.

## Methods

**Library design and cloning**. Custom Python scripts were written for the pairwise library design. Initially, we generated a random library of 19–24 nt spacer sequences. To minimize undesired Cas9 targeting outside the lentivirally-integrated pairwise library cassette, the sgRNA sequences were then computationally optimized to be highly orthogonal to the human reference genome by filtering the list of candidate spacers against the hg19 assembly. A 5′ G was held constant for every unique sgRNA spacer for reliable U6-driven expression. The (PAM) was held constant at 5′-CAGGGT-3′. The number of single-mismatch, double-mismatch, DNA bulge, RNA bulge, and control guides that were generated for each unique sgRNA spacer is described in Supplementary Table 1.

Hamming codes were generated using a modified version of a Python script available on Github (https://github.com/mdshw5/hamstring). We modified the code to increase the barcode length to 15-bp, expanding the number of available barcodes so as to cover the whole library. These barcodes are composed of ten data bases, four checksum bases, and one parity base. We excluded homopolymers of >3 nt and filtered for GC content from 30% to 70%. This resulted in 812,547 barcodes, which were sub-sampled to 93,000, enabling an increased edit-distance greater than 2.

The library was synthesized as a pool of single-stranded 135–142 nt oligonucleotides by CustomArray. Variable oligo length is due to the varied spacer lengths. In order to accommodate constraints of synthesis length, oligo synthesis did not include the full length sgRNA tail, but instead included a short BsmBI Type IIS tracrRNA cloning site in between the spacer and target sequences.

First, the library oligo was PCR amplified and cloned by Gibson Assembly (NEB) into the pairwise library lentiviral backbone, which included a U6 promoter

for sgRNA expression and a puromycin resistance gene (Supplementary Data 1 and Supplementary Data 2). The plasmid was then electroporated into Endura ElectroCompetent cells (Biorad). To maintain library complexity, the transformed cells were plated on large 245 × 245 mm LB plates (Teknova) and colony density was estimated by serially diluting and spreading the transformed cells on LB plates. Colonies were quantified with an online image analysis tool (Benchling).

This pre-tracrRNA pairwise library was sequenced by MiSeq to verify its representation and rate of synthesis errors. The tracrRNA was synthesized as a PAGE-purified ultramer (Integrated DNA Technologies), with an additional 8 nt 5′-NNNNNNNN-3′ rBC at the 3′ end to identify independent DNA molecules representing each library member. The tracrRNA-rBC oligo was PCR amplified and ligated into the BsmBI cloning site of the pre-tracrRNA. The resulting plasmid pool was also subjected to deep sequencing by MiSeq to verify library member representation and synthesis quality. The library sequences are available (Supplementary Data 3).

**Cell culture**. HEK293T (ATCC, CRL-3216) and HEK293FT (Life Technologies, catalog #R700-07) cells were cultured in Dulbecco's modified eagle medium (DMEM), supplemented with 10% fetal bovine serum (FBS) and 1% penicillin-streptomyocin (D10). Cells were maintained in T225 flasks while screening. The antibiotic selections used 10 μg/ml blasticidin or 0.5 μg/ml puromycin. HEK293-GFP (GenTarget, catalog #SC001) cells were maintained in DMEM, supplemented with 10% FBS, 5% penicillin-streptomycin, and 2 mM Glutamax. All HEK293 were kept at 37 °C in a 5% $CO_2$ incubator. These cell lines were used due to their efficient transfection and lentiviral transduction, which make them standard cell lines for CRISPR-Cas9 characterization in eukaryotic cells. These cell lines were purchased directly from the manufacturers but not otherwise authenticated.

**Lentiviral production and titering**. To package lentivirus, the library plasmid was co-transfected with pMD2.G and psPAX2 into 120 million HEK293FT cells, using Lipofectamine 2000 (Thermo Fisher) in 10 T-225 flasks (Sigma). After 72 h, the supernatant was harvested and concentrated according to the LentiX Concentrator protocol, then stored at −80 °C. In order to titer this lentiviral preparation, we spinfected 3 million HEK293T cells per well with 0, 40, 80, 120, 160, or 200 μl of the lentiviral supernatant in 2 ml D10 media supplemented with 8 μg/ml polybrene (Sigma) in 12-well plates (Sigma). The plates were spun at 1000 × *g* for 2 h at 37 °C. After spinning, 2 ml fresh D10 media was added to each well and the cells were maintained at 37 °C for 24 h. The cells were then dissociated with TrypLE, suspended in D10, and counted with a TC20 Automated Cell Counter (Biorad). Then, 2500 cells per well were plated in a black TC-treated 96-well plate with clear well bottoms (Sigma). For each dose, four wells then underwent puromycin selection while four wells continued growth in D10 media. Media was refreshed after 48 h. After 96 h, survival of the selected cells in comparison to the unselected cells was measured by CellTiter-Glo (Promega) and used to calculate the lentivirus MOI. The SaCas9 lentivirus was similarly prepared and titered, using blasticidin selection.

**Pairwise library arrayed pilot**. As a pilot study, we generated five lentiviral vectors with a shared sgRNA targeting five different target sites, which were mismatched at varied positions to examine specificity. These were synthesized by IDT, cloned into the library lentiviral backbone, and verified by Sanger sequencing. Each construct was individually packaged into lentivirus and titered. HEK293T were transduced by spinfection with the SaCas9 lentivirus and underwent 6 days of blasticidin selection. Then, wells of 3 million cells each were spinfected with the guide-target lentiviruses, in duplicate, and subjected to puromycin selection for 7 days. Cells were harvested every 24 h. Genomic DNA was extracted and the target sites were sequenced by MiSeq to measure indel rates, using the same computational pipeline as for the pooled screen.

**Pairwise library pooled screening workflow**. Starting from a library of 88,692 members and 27% error-free reads in the post-tracrRNA plasmid pool; we spinfected 340 million HEK293T cells with the library lentivirus at a low multiplicity of infection (MOI = 0.3) to achieve a desired representation of 300×. Twenty-four hours after spinfection, we began 5 days of puromycin selection. The cells recovered in antibiotic-free D10 media for 1 day. Then, 340 million cells were harvested for the Day 0 timepoint and 820 million cells were spinfected with the SaCas9 lentivirus at an MOI = 0.4 to achieve 1000× representation. The cells recovered in antibiotic-free D10 media for 1 day. They were maintained for 7 days under blasticidin selection and then maintained without further selection for an additional 6 days. The biological replicate of the screen was performed independently on a different day, starting from the initial library spinfection.

**Library preparation and sequencing**. Genomic DNA was purified from cell pellets of over 340 million cells with a Quick-gDNA Midiprep kit (Zymo), according to the manufacturer's instructions. The library-cassette regions from the entire DNA pool were then PCR amplified with primers designed to target the U6 promoter and the constant sequence downstream of the target site. These primers include Illumina sequencing adapters as extensions (P5 on the forward primer and P7 on the reverse primer) (Supplementary Data 1). There were 12 forward primers

and 8 reverse primers, which differed in length due to a stagger sequence. The stagger ensures a diversity of base calls in each cycle of sequencing. For each sample, 96 PCRs were performed with: 20 μg gDNA, 25 μl NEBNext Master Mix (NEB), 0.5 μl Q5 polymerase (NEB), 1 μl DMSO, 1 μl MgCl₂ (25 mM), and 0.5 μM forward and reverse primers, in 50 μl reactions. The thermocycling protocol was: 30 s at 98 ⁰C, followed by 18 cycles of 98 ⁰C for 10 s, 60 ⁰C for 30 s, and 72 ⁰C for 30 s, then a final 5 min extension at 72 ⁰C. Then, the 96 reactions were pooled. The desired PCR products were then gel extracted and quantified by Qubit. These libraries were subjected to 2 × 150 bp paired end sequencing on a HiSeq2500 (Genewiz), with two lanes per sample.

**Computational analysis**. Indel rates were computed with a Python script[27]. The indel caller discarded reads with substitutions in critical regions, which may arise from sequencing or PCR errors. Error rates were estimated by comparing the constant regions of the library cassette against reference sequences. The pairwise library pooled screen was analyzed with two pipelines. The first pipeline took the library design file and the Day 0 sequencing reads as inputs and outputted a whitelist of validated rBCs, which were associated with error-free library cassettes. The second pipeline took this rBC list and the Day 3 or 14 sequencing reads as inputs and outputted the indel rate for each guide-target pair, computed from the subset of reads bearing whitelisted rBCs. This process minimizes false positive indels that may arise from mutations in the target site during the synthesis, cloning, or lentiviral packaging. Off:on-target ratios are an indel rate normalized by the indel rate resulting from the perfectly matched guide of equivalent length. We computed this ratio for guide groups with on-target activity >2%. The script was written in Python and run on an Amazon Web Services EC2 instance. Statistical analyses were performed in Graphpad Prism. Two-sided tests were used in all cases.

**GFP reporter of mismatched sgRNA activity**. Wild-type and mismatched sgRNA were generated by PCR and transfected as amplicons containing U6 promoter, spacer sequence, and tracrRNA scaffold. Singly mismatched sgRNA were designed by swapping T and A, or G and C, at each position. HEK293-GFP were seeded at a density of 100,000 cells/well in 24-well plates. After 24 h, cells were transfected with 250 ng of gRNA amplicon and 750 ng of wild-type SaCas9 plasmid (pAF003). All transfections were performed in duplicate with Lipofectamine 3000 (Life Technologies). At three and a half days post-transfection, cells had media removed and were washed with 0.5 mL of phosphate-buffered saline (PBS). Two hundred microliters of trypsin was added to the cells and they were incubated at 37 °C, 5% CO₂ for 5 min. Trypsinization was halted by adding 0.5 mL of complete media to each well. Cells were collected and transferred to 1.5 mL tubes, spun down at 1000 × g for 7 min, washed with 1.0 mL fluorescence-activated cell sorting (FACS) buffer (PBS with 3% FBS), spun down again, and resuspended in 200 μl FACS buffer. Cells were then analyzed with a BD Accuri C6 flow cytometer.

**Northern blot**. HEK293T cells were transfected with plasmids expressing sgRNA of varying lengths (18, 20, 22, and 24 nt) with a U6 promoter. Samples were co-transfected with a SaCas9 plasmid or the corresponding empty vector. After 3 days, sgRNA were purified from cell pellets with a mirVana kit for small RNA isolation (Thermo Fisher). RNAs were heated to 95 °C for 5 min before loading on 8% denaturing polyacrylamide gels (SequaGel, National Diagnostics). Afterwards, RNA was transferred to a Hybond N+ membrane (GE Healthcare) and crosslinked with Stratagene UV Crosslinker (Stratagene). Probes were labeled with (gamma-³²P) ATP (PerkinElmer) with T4 polynucleotide kinase (New England Biolabs). After washing, membrane was exposed to phosphor screen for 1 h and scanned with a phosphorimager (Typhoon). The sgRNA expression ratio was quantified based on the intensity of the bands in the image using ImageJ analysis software.

**Model of SaCas9 specificity**. Parameter values for the nonlinear model were derived using Hamiltonian Monte Carlo sampling as implemented in the Rstan package. The default No U-Turn Sampler was used to draw 1500 samples across eight independent chains, with the first 500 samples in each chain being discarded as part of the warmup phase. Convergence of the posterior distributions for the parameters was verified by calculating the R-hat statistic and verifying that values for model parameters were <1.1.

The statistical model is implemented as follows. The indel rate $I_{j,M}$ observed for a guide $j$ targeting a sequence with mismatches $M$ is calculated by assuming that the on-target activity of the guide $g_j$ is decreased by additive penalties incurred for each mismatch ($\Delta\Delta G_k$) and that the calculated indel rate is subject to independent Gaussian errors $\epsilon_{j,M}$.

$$I_{j,M} = g_j \exp\left(\sum_{k \in M} -\Delta\Delta G_k\right) + \epsilon_{j,M} \qquad (1)$$

The penalty terms $\Delta\Delta G_k$ are restricted to being non-negative and regularized by enforcing a prior distribution that is exponential with mean $\beta = 1$. The error terms $\epsilon_{j,M}$ are normally distributed with mean 0 and standard deviation 0.1 (as estimated from comparisons across replicates). For ease of interpretation, the same model can

be reparametrized so that the penalties are multiplicative, as follows:

$$I_{j,M} = g_j \prod_{k \in M} f(\text{pos}(k), \text{type}(k)) + \epsilon_{j,M} \qquad (2)$$

In this case, the penalty on guide activity as a function of the mismatch position and type can be represented as a value between 0 and 1. Parameter values were independently derived for each guide length by separating the data into disjoint training sets, leading to a separate mismatch effect matrix $f_L$ for each value of guide length $L$ assayed.

This score fit the observations well for most guides in our dataset, except for guides with very low on-target activity (Supplementary Fig. 5B, C). This may be explained by a low signal-to-noise ratio for these ineffective guides.

**Code availability**. The code for the analysis, modeling, and specificity score are available on the Editas Medicine Github (https://github.com/editasmedicinedev/pairwise-library-screen). They are available under the BSD 3-Clause Clear License.

**Data availability**. The datasets generated during the current study are available in the NCBI SRA repository (SRP147992).

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

## Acknowledgements

We would like to thank Team Editas for helpful discussions and support.

## Author contributions

P.D.H. devised the concept. J.T., J.S.G., and P.D.H. designed the library with input from O.O.A. J.T., N.H., and P.D.H. carried out screen. A.E.F. performed G.F.P. knockout validation. X.W. performed Northern blots. J.S.G. contributed indel detection code. J.T. and P.D.H. wrote screen analysis code and performed data analysis. L.A.B. developed the specificity score. J.T., L.A.B., V.E.M., C.J.W., and P.D.H. interpreted results and wrote the manuscript with help from all authors.

## Additional information

**Competing interests:** J.T., L.A.B., N.H., A.E.F., J.S.G., V.E.M., C.J.W., and P.D.H. are employees, consultants, or former employees of Editas Medicine. X.W. and O.O.A. declare no competing interests.

