## [Peer Review File · Nature Communications]

Reviewers' comments:

Reviewer #1 (Remarks to the Author):

Tycho et al. describe a high-throughput screening approach to identify variables affecting the genome editing efficiency and specificity of the *Staphylococcus aureus* CRISPR/Cas9 system. To this end, they profile the genome editing activity of SaCas9 across a large repertoire of sgRNAs paired with matched or mismatched targets. In particular, the authors generated a library of lentiviral vectors each containing a pair of sgRNA-target linked to two barcodes: a hamming barcode, identifying the original sgRNA-target member in the library, and a randomized barcode (rBC), each identifying a different cell integrant for any given sgRNA-target pair. Upon library transduction in HEK-293T cells, and before Cas9 delivery, the rBC was exploited to discard both sgRNA-target pair showing indels in the target prior to Cas9 delivery and library members with low cellular representation. By exploiting this approach, the authors were able to reliably compare the on- and off-target editing efficiency among sgRNAs with spacer length ranging from 19 to 24 nucleotides (nts). These studies showed that the 21 nts sgRNA displayed the highest on-target efficiency. This was, however, associated to the highest off-target occurrence. On the other hand, 22 nts spacer offered the best balance between efficiency and specificity, while the standard 20 nts spacer has better specificity at cost of less efficiency. Of note, and in line with previous studies, in terms of off-target likelihood, the authors showed that single (and to a much lesser extent double) mismatches in the target are well tolerated by the sgRNA, especially in the PAM-distal region, while they are poorly tolerated in the PAM-proximal region. On the other hand, buldges in the target or in the sgRNA are minimally tolerated at PAM-proximal regions, while PAM-distal buldges are near-completely tolerated. Furthermore, by training their dataset, the authors generated a scoring algorithm that was able to predict SaCas9 specificity in a GFP mismatch assay (generated in this work) and rank endogenous off-target sites retrieved by previously reported GUIDE-seq data. Overall, the manuscript is well written, the study is scientifically sound, and the methodology used is largely appropriate. The experimental strategy, although previously anticipated for Cpf1 in other studies but implemented here by the addition of two barcodes, might help designing more efficient and specific sgRNAs for SaCas9, a relevant point towards clinical application of genome editing. Finally, the algorithm here developed might be of broad utility to predict and rank off-target sites for a given sgRNA of interest. However, to strengthen their findings, the authors should address the following points.

1. The authors should assess whether their prediction model is valid also in other cell types.
2. With the exception of the pilot experiment shown in Figure S1, the authors perform all the off: on-target analyses at day 3 post SaCas9 delivery, that is in a non-saturating condition. However, in Figure S1, they show that both on- and off-target activity increase from day 3 to day 6. Therefore, the authors should perform their screening assay also at day 6, and address if: a) the 21 nts spacer remains the best performing one in terms of on-target activity; b) the 20 nts spacer is still the most specific, or if differences are abrogated at saturation; c) RNA and DNA buldges are still resistant to Cas9 activity. It would be interesting to perform similar experiments also by transiently transfecting RNPs, a strategy used to decrease off-target activity of Cas9.
3. In the study the authors compared sgRNA with fixed proximal spacers and varying distal nts length, showing that the 20 nts spacer has the best specificity, while the 22 nts has the best activity. Is this finding holding true also when the scaffold proximal nts are kept fixed, while the scaffold distal vary?
4. The authors should specify in the supplementary materials the sequence of the construct including the target and the hamming code and in the main text the distance between the PAM and the Hamming code / NGS primer. If the distance is not long enough, DNA repair events consisting in medium-long deletions spreading to the Hamming code/NGS primer may be lost in the analyses. This could explain the suboptimal on target activity shown in Figure S1, not reaching 100% efficiency even at late time points. In this case both on- and off-target activity would be underestimated by this approach. The authors should address this point.
5. The results shown in Figure S3a suggest that a 5'G improves sgRNA stability. However, a

reporter for transfection efficiency (i.e. a GFP expression cassette inserted in the same plasmid encoding for the sgRNAs) should be included to confirm similar transfection efficiency between the sgRNAs.

Reviewer #2 (Remarks to the Author):

In this manuscript, Tycko and colleagues utilized a high-throughput screening approach to characterize SaCas9 activity and specificity in human cells. Using a large repertoire of sgRNA-target combinations (matched or mismatched) for pairwise library screening, the authors characterized the guide-target activity relationships as a function of guide length, number of mismatches and types of mismatches in integrated genomic target sites in cells. The authors train a computational model on these data to create an algorithm to predict the SaCas9 activity as a function of guide mismatches and then prospectively test this model and demonstrate its utility.

Overall this manuscript demonstrates the value of an in depth screening approach to define the most favorable parameters for optimal nuclease activity and specificity in human cells. Interestingly, the authors find 21-nucleotide spacers are both the most active guide length and have the highest tolerance for mismatches, compared to shorter or longer guide lengths. This observation will be useful for the scientific community and for clinical translation of SaCas9 for targeted genome engineering.

I have some concerns about aspects of this manuscript:

1. Important issue: The authors investigated 73 different guide groups, but for the majority of the guides the activity is quite low (median for all groups is < 10%) and for most of the guide lengths the on-target activity appears to be less than 4% (less than 1 in 25 integrated sequences for the on-target site are mutated). Based on their Whisker plot in Figure 1E for many of the groups the lower quartile has on-target activities that are less than 1% (<1 in 100 mutated sequences). The authors are then representing the activity profile of mismatched DNA target sites as ratio of off-target to on-target ratio. However, their library depth is at best 1000x (1000 genome copies of each guide-target combination) based on their calculations within the methods section. This would not appear to allow the necessary depth of genomes present to accurately estimate mismatch editing rates for many of their guides (i.e. if there are only 10 mutant alleles for the on-target site, how to accurately estimate a rate for an "off-target" guide/target combination). This challenge in mutation rate estimation does not include the problems of the background error rate for Illumina sequencing, which can be high for some sequences (0.1 to 1%). The authors should clarify how they addressed these issues in the manuscript because for some mismatches (in particular the double-mismatches or DNA/RNA bulges), they may not allow accurate estimation of activity to spacer length (non-21 nt spacer have lower activity and thus a lower dynamic range that can be sampled).

2. The activities for the on targets show large variations for different spacer length (Figure 1E), and the off:on-target ratios also show large variations in activity (Figure 1F). Is there a correlation between the on-target activity and off-target activity in the dataset? Is there a dichotomy in the sgRNAs population that is leading to the large variations in both on-target activity and in off:on-target activity ratios?

Minor comments:

- Table 1 listed in the text actually appears to be Supplementary Table 1 (or Table 1 is missing.)

Nature Communications Review: Executive Summary

We would like to thank the editor and reviewers for the helpful questions and constructive feedback that have helped us to significantly improve this manuscript. In particular, we were pleased that the reviewers found our work “well written”, “scientifically sound”, “useful for the scientific community and for clinical translation”, and that the algorithm we developed to be “of broad utility to predict and rank off-target sites”.

To summarize the major additions to the manuscript due to the reviewers' suggestions:

1. We now include a day 14 time-point for the pairwise library specificity screen that addresses comments about the consistency of our specificity analysis. In particular, do our findings hold true when there is higher indel activity? We have included the on-target, 1-mismatch, and 2-mismatch data from day 14 in a new **Fig. S2**, while the RNA/DNA bulge data from day 14 are now described in **Fig. S3**. In short, we draw the same conclusions about SaCas9 specificity between day 3 and day 14. For example, we show that editing efficiency as a function of sgRNA spacer length is consistent even with higher indel activity.
2. We include details in the revised text and methods that clarify the design and analysis of our pairwise library screen. In particular, we provide:
 - a. Schematic of the library cassette including nucleotide distances between critical components such as the target site, Hamming code, and primer binding site (**Fig. S1A**)
 - b. Important details about the analysis code are now included in the written text. In particular, guide-target pairs with <2% on-target indel activity were omitted from our analysis of off:on-target ratios in order to avoid sequencing sensitivity issues. Also, we used an indel caller algorithm that properly detects deletions but discards reads with substitutions, to avoid false positives that may come from sequencing and PCR errors.

We also thank the reviewers for leading us to clarify that we validated our SaCas9 specificity prediction model using GUIDE-seq data **from a different cell type**, U2OS. This important detail had previously been omitted.

The reviewers raised very interesting questions about the design of the screen (How to shorten/lengthen spacers?) and the technical challenges associated with the method (How to handle low-activity guides? How to handle sequencing errors?). Please find in-depth, point-by-point responses attached.

We believe this revised manuscript is much stronger due to these insightful reviews and appreciate the reviewers' time and expertise.

Reviewer #1 (Remarks to the Author):

1. The authors should assess whether their prediction model is valid also in other cell types.

This is a great point and we agree that it is important to validate the findings in a different cell type. To that end, we used our specificity model to predict GUIDE-seq off-target measurements in U2OS cells, and show the results in **Fig 3**. We saw strong correspondence with the predicted and observed off-target edits, with a Spearman correlation of 0.82. Previously, we had failed to clarify the cell type of that experiment. We have corrected this omission in the appropriate figure legend and the Results section of the manuscript.
2. With the exception of the pilot experiment shown in Figure S1, the authors perform all the off:on-target analyses at day 3 post SaCas9 delivery, that is in a non-saturating condition. However, in Figure S1, they show that both on- and off-target activity increase from day 3 to day 6. Therefore, the authors should perform their screening assay also at day 6, and address if:
 - a) the 21 nts spacer remains the best performing one in terms of on-target activity;
 - b) the 20 nts spacer is still the most specific, or if differences are abrogated at saturation;
 - c) RNA and DNA bulges are still resistant to Cas9 activity.

Thank you for this suggestion. We agree that the comparison of a non-saturated condition with a late time point would be useful in order to understand if these findings are applicable in different genome-editing scenarios. To that end, we extracted genomic DNA from the same screen 14 days after SaCas9 delivery and performed similar analyses. We observe that:

- A) The 21 and 22 nt spacers retain the highest levels of on-target activity as in the day 3 time point. All spacer lengths had increased activity relative to day 3, as expected. This data is now included in **Fig. S2A** and the results stated in the main text.

B) We do not see evidence of abrogated differences between the spacer lengths in their mismatch tolerance at Day 14. As now stated in the Results: “The 20 nt spacer was significantly less tolerant of single mismatches than the 21 nt or 22 nt spacer at both Day 3 and Day 14 ($p < 0.01$, Dunn’s multiple comparisons test, **Fig. S2B**).” The trend of single mismatch tolerance across spacer lengths and mismatch positions is very similar, with the 21 nt spacer being the most tolerant of mismatches (**Fig. S2B**). The 21 nt spacer length was also significantly more tolerant of double mismatches than any other length at both time points, and this is now stated in the Results. The 20 nt spacer length was significantly less tolerant of double mismatches than any longer spacer at both time points ($p < 0.0001$, Dunn’s multiple comparisons test) .
C) Overall, the pattern of bulge tolerance is similar at Day 14, wherein the PAM-proximal bulges result in <25% off:on-target ratio and the bulges up to 4-nt from the PAM-distal end are nearly fully tolerated. We do detect an increase in tolerance of some bulges after 14 days, as is now shown in **Fig. S3**.

2.1 It would be interesting to perform similar experiments also by transiently transfecting RNPs, a strategy used to decrease off-target activity of Cas9.

We appreciate that RNP transfection is an important delivery modality with potential specificity advantages. In particular, the RNP strategy is underutilized with the SaCas9 ortholog as the majority of RNP work has been done with SpCas9. We note that a primary advantage of SaCas9 for genome editing is its short coding sequence for size-constrained viral vectors, which is not relevant for RNP-based delivery. However, understanding RNP-mediated editing specificity would certainly be useful in studies that multiplex two orthogonal RNPs such as SaCas9 and SpCas9.

In this study, we sought to control for delivery variability by introducing a single copy of the guide-target cassette per cell by using low-MOI lentiviral transduction. This strategy enables delivery of a pool of guide-target cassettes to test large numbers of mismatched sites in a single experiment, which would not be straightforward with RNP approaches. As such, we believe our approach is better suited to isolate the variable of sequence-intrinsic effects to study their impact on specificity, and to generate a model that is broadly applicable for predicting SaCas9 off-target effects.

In addition, many of our authors have contributed to an independent study of SaCas9 RNP specificity that is currently in the peer review process. We believe these two complementary manuscripts will together provide a comprehensive understanding of SaCas9 specificity across different delivery methods.

3. In the study the authors compared sgRNA with fixed proximal spacers and varying distal nts length, showing that the 20 nts spacer has the best specificity, while the 22 nts has the best activity. Is this finding holding true also when the scaffold proximal nts are kept fixed, while the scaffold distal vary?

While our study was not designed to specifically answer this question, this is certainly an interesting consideration. To rephrase the reviewer’s comment, we interpret this question as: “What would happen if spacer length changed, while keeping the PAM-distal bases the same?” Put another way, “What would be the consequence of adding or removing the PAM-proximal bases?” There are two primary reasons we chose to instead add/remove PAM-distal bases and not the PAM-proximal bases:

1. First, “how would this affect our measurement of the **on-target** activity?” It is not possible in a ‘real’ genome with fixed sequence to change the spacer length by varying the ‘proximal nts’ while targeting the same site. The exception is when there are two overlapping PAMs. In other words, to change a 20-nt spacer to a 21-nt spacer while holding the PAM-distal nts fixed, one must add a single PAM-proximal base to effectively slide the PAM out by one base. The SaCas9 PAM (NNGRRT) is not amenable to sliding one nt in either direction, unless one slides ‘up’ one nt to utilize the alternative PAM ‘NNGGRR’, which is weaker but still usable. It is rare in a guide design scenario for genome editing to make a spacer length choice that slides the PAM instead of simply extending the spacer on its 5’ end, because there is rarely such a new PAM available. In sum, our chosen format answers the more relevant question for guide design for most genome editing scenarios.
2. Second, “how would this affect our measurement of the **off-target** activity?” SpCas9 is known to have position-dependent mismatch tolerance, and we have now shown this for SaCas9 too. In this alternative scenario, one would be changing the length of the spacer, and also the PAM-relative position of the mismatches. This would confound the ability to determine if the spacer length modulates a change in the off-/on-target activity ratio, because any change in specificity could also be due to the change of mismatch position. For example, in this alternative scenario, a 21-nt spacer with a mismatch at position 3 would be compared with a 22-nt spacer with the mismatch now pushed out to position 4 – this comparison would be confounded by the change in mismatch position relative to the PAM.

4. The authors should specify in the supplementary materials the sequence of the construct including the target and the hamming code and in the main text the distance between the PAM and the Hamming code / NGS primer. If the distance is not long enough, DNA repair events consisting in medium-long deletions spreading to the Hamming code/NGS primer may be lost in the analyses. This could explain the suboptimal on-target activity shown in Figure S1, not reaching 100% efficiency even at late time points. In this case both on- and off-target activity would be underestimated by this approach. The authors should address this point.

Thank you for raising this important point; we apologize for not being clear. The sequence of the cassette with an example guide-target pair is in **Supplementary Table 3**, plasmid pPDH002. We have now added a schematic with sequence and distance measures to **Fig. S1**.

The Hamming code starts 16-bp downstream from the predicted cut site. This is now included in the main text. The closest NGS primer starts 31-bp downstream from the predicted cut site.

We found the Hamming code in the great majority of sequencing reads, which suggests that very few true indel events were missed due to Hamming code disruption. Across sequencing runs, we found a match to the Hamming code in 83% of reads, on average. After using our error-correcting algorithm to retrieve the missing Hamming codes, we were able to match 23% of the remaining reads to a Hamming code, on average. These details are now included in the main text.

It is possible that some long deletions (>16-bp) disrupt the Hamming code to the extent that error-correction fails, and these could account for some portion of the remaining 13% of reads that have no recognized Hamming code. Furthermore, longer deletions >31-bp may disrupt the library PCR and therefore be excluded from the sequenced library. We assume these potential PCR-disrupting events are quite rare, as they should be even rarer than the potential Hamming code disruptions. Studies of Cas9-induced indel distribution typically show that the indel distribution looks semi-normal centered around the cut site (e.g. Paquet et al. Nature 2016). Furthermore, these types of errors are likely to be distributed across the library members so they would not significantly affect our conclusions about the rules of SaCas9 specificity.

We also do not expect that they would predominantly affect a certain category of the library and thus skew the calculation of the "Off:on-target ratio". In future studies, it would certainly be interesting to use the pairwise library concept we describe here to specifically study the DNA repair outcomes of different guides and targets. These data would help us better understand if certain sites are more prone to long deletions, as has been suggested in some microhomology studies (Bae et al. Nature Methods 2014). An additional plausible reason for the <100% efficiency in **Fig. S1** is epigenetic silencing of the lenti-Cas9 transgene in the HEK293T cells (likely after blasticidin selection is removed); this has been anecdotally reported for this cell line by several labs.

Overall, for the purposes of studying genome editing specificity, we think it is encouraging that the screen data was used to train a model that validates with strong correlation on two orthogonal specificity assays, neither of which would suffer from the same potential issues.

5. The results shown in Figure S3a suggest that a 5'G improves sgRNA stability. However, a reporter for transfection efficiency (i.e. a GFP expression cassette inserted in the same plasmid encoding for the sgRNAs) should be included to confirm similar transfection efficiency between the sgRNAs.

This is an interesting point for further study, given the widespread use of the 5'G for guide RNA expression. Here, we do not seek to claim that the 5'G improves sgRNA stability. Rather, we were interested to determine if the 5'G affects the length of the sgRNA transcript and concluded that the G was not necessary for full-length expression.

We note that our U1 normalization for total RNA (**Fig. S4**) demonstrates that the sgRNAs are expressed at similar levels given identical transfection conditions, and that expression level effects do not confound our measurement of transcript length. Regardless of this result, we did use the 5'G for all guides tested in the screen and in the orthogonal assays used to validate the screen, as is standard in the field.

Reviewer #2 (Remarks to the Author):

I have some concerns about aspects of this manuscript:

1. Important issue: The authors investigated 73 different guide groups, but for the majority of the guides the activity is quite low (median for all groups is < 10%) and for most of the guide lengths the on-target activity appears to be less than 4% (less than 1 in 25 integrated sequences for the on-target site are mutated). Based on their Whisker plot in

Figure 1E for many of the groups the lower quartile has on-target activities that are less than 1% (<1 in 100 mutated sequences). The authors are then representing the activity profile of mismatched DNA target sites as ratio of off-target to on-target ratio. However, their library depth is at best 1000x (1000 genome copies of each guide-target combination) based on their calculations within the methods section. This would not appear to allow the necessary depth of genomes present to accurately estimate mismatch editing rates for many of their guides (i.e. if there are only 10 mutant alleles for the on-target site, how to accurately estimate a rate for an “off-target” guide/target combination).

Thank you for raising this important point. We have tried to break down this issue into several questions and address them below:

- First, “why is the activity low?” We agree that the SaCas9 activity is relatively low. This is intentional using a low MOI (0.3) of SaCas9 expressing lentivirus to ensure most cells get just one copy of SaCas9. However, our protocol results in significant portion of cells, ~70%, to not be transduced by SaCas9 lentivirus and, as expected, this is reflected in the results. Blasticidin selection for the lenti-SaCas9-transduced cells is not complete by Day 3.
- Second, “how do we handle the guide groups with low on-target activity?” We apologize for not previously being clear on this important issue. During analysis, we set a threshold such that the on-target activity must be >2% in order for Off:On-target ratios to be computed and included for that group of guide-target pairs. So, there are data points in **Fig. 1E** that are below 2% and are then filtered out for subsequent specificity analyses in the later figures. This important detail is now included in the Methods section. While this lowers our ‘N’ for those analyses, we agreed that measurement accuracy would decline as the indel rates approach the sequencing error rates, and think this filter is important. N is listed in every figure legend and was still orders of magnitude higher than previous studies, providing sufficient power to demonstrate the statistical significance of the effects we observe.
- Third, “what sequencing depth was achieved?” The reviewer correctly states that we aim for 1000x representation of each library member in the pool of cells. The average achieved sequencing depth for all library members included in the specificity analyses is 2,759 reads distributed across 113 unique lentiviral integrations (i.e. RBCs, range = 20 - 1,898). From another perspective, the sequencing depth relevant to a given measurement depends on the analysis done, as our figures are generated from aggregates of large numbers of guide-target pairs. For example, in **Fig. 2A** the black triangle at position 5 is an aggregate of all 23-nt spacer guide-target pairs (n = 229) that have a single mismatch at position 5. That measurement is the result of analyzing 439,840 sequencing reads, that were distributed across 29,496 RBCs. On average, each circle in that figure is the result of 563,641 sequencing reads distributed across 35,666 RBCs. The rest of **Fig. 2** is similarly aggregated data.
- Fourth, we do not make claims about the off:on-target ratio measurement of an individual guide-target pair in this manuscript. In the future, researchers might be interested in the characteristics of particular guide-target pairs as opposed to general principles of SaCas9 specificity. Then, one would modify this method accordingly by using a smaller library, or more cells, or more stringent RBC filtering to include only the most accurate measurements.
- Fifth, “does the low activity qualitatively affect the findings about SaCas9 specificity?” To address this point, and complementary questions from Reviewer 1, we are adding new data and figures derived from genomic DNA extracted from the same screen 14 days after SaCas9 delivery. We find:
 - A) The effect of spacer length on on-target activity is consistent. The 21 and 22 nt spacers remain the most active on-target, although all spacer lengths had increased activity over that time period, as expected. This data is now included in **Fig. S2A** and the result stated in the main text.
 - B) The effect of spacer length on mismatch tolerance is consistent across time points. As now stated in the Results: “The 20 nt spacer was significantly less tolerant of single mismatches than the 21 nt or 22 nt spacer at both Day 3 and Day 14 ($p < 0.01$, Dunn’s multiple comparisons test, **Fig. S2B**.)” The trend of single mismatch tolerance across spacer lengths and mismatch positions is very similar, with the 21 nt spacer being the most tolerant of mismatches (**Fig. S2B**). The 21 nt spacer length was also significantly more tolerant of double mismatches than any other length at both time points, and this is now stated in the Results. The 20 nt spacer length was significantly less tolerant of double mismatches than any longer spacer at both time points ($p < 0.0001$, Dunn’s multiple comparisons test) .
 - C) Overall, the pattern of bulge tolerance is similar at Day 14, wherein the PAM-proximal bulges result in <25% off:on-target ratio and the bulges up to 4-nt from the PAM-distal end are nearly fully tolerated. We do detect an increase in tolerance of some bulges after 14 days, as is now shown in **Fig. S3**.

This challenge in mutation rate estimation does not include the problems of the background error rate for Illumina sequencing, which can be high for some sequences (0.1 to 1%). The authors should clarify how they addressed these issues in the manuscript because for some mismatches (in particular the double-mismatches or DNA/RNA

bulges), they may not allow accurate estimation of activity to spacer length (non-21 nt spacer have lower activity and thus a lower dynamic range that can be sampled).

Thank you for bringing up this very relevant issue. We sought to address the challenge of sequencing errors in several ways:

- Sequencing errors mostly result in substitutions or indels at homopolymers. We used an indel calling algorithm that specifically treats substitutions as a type of error and discards those rare reads that do have substitutions in the target site or other critical regions of the amplicon (e.g. PAM). These sequencing errors thus do not get called as 'false positive' indels. This important detail is now stated in the Methods section. Additionally, we avoided homopolymers in the design of target sites.
- As described above, we sequenced to a high depth, which dilutes out the effect of 'mutations' that are randomly distributed across the sequencing amplicon.
- One could imagine that for some reason certain library members are particularly prone to recurrent errors in a non-random position. However, the effect of these would also be diluted out in our analyses that aggregate large numbers of guide-target pairs. Also, such library members would be filtered out in the initial step of 'whitelisting' only RBCs associated with error-free library cassettes at Day 0.
- One could also imagine sequencing errors in the barcode affecting the analysis - our use of error-correcting Hamming codes allows us to algorithmically correct such errors and properly associate barcodes with library members.
- For these reasons, and because of validation in orthogonal assays, we do not think sequencing errors have significantly confounded our findings.

2. The activities for the on targets show large variations for different spacer length (Figure 1E), and the off:on-target ratios also show large variations in activity (Figure 1F). Is there a correlation between the on-target activity and off-target activity in the dataset?

Thank you for this interesting question. We see a weak but significant correlation ($R^2 = 0.16$, $p < .0001$) between the on-target activity and off-target activity of the guide-target pairs with single mismatches. The correlation is stronger ($R^2 = 0.35$) if we consider only PAM-distal mismatches, as expected. But, we see no correlation when we consider guide-target pairs with 2 mismatches or bulges. Overall, there is little correlation because the vast majority of the off-target sites are double-mismatch sites that are minimally edited by SaCas9.

Is there a dichotomy in the sgRNAs population that is leading to the large variations in both on-target activity and in off:on-target activity ratios?

This is an important question and indeed one of the motivations of this study. Variability in guide activity is an important problem in the field that high-throughput methods in cells are well-positioned to resolve by adding statistical power. While overall variability in sgRNA activity is relatively high (Fig 1E), with the large dataset we are able to isolate sequence-dependent effects. For example, we are able to show how spacer length affects off:on-target activity when controlling for mismatch position and identity (Fig 2A). We do not observe a single dichotomy in the sgRNA population that drives variable specificity, but rather an integrated function of at least 4 parameters (mismatch position, identity, number, and spacer length).

We also agree that there is a need to understand determinants of on-target variability. Many genome editing experiments still begin with experimental screening of sgRNA to find optimal spacer sequences, despite some data-driven efficiency models for SpCas9 (e.g. Doench et al. Nature Biotechnology 2016). Here, our specificity screen included 653 on-target guide-target pairs. We did not observe a dichotomy in the sgRNA population that explains variable on-target activity. However, this on-target data set is >30x larger than previous SaCas9 efficiency studies, and powered an analysis that indicated the maximally efficient on-target spacer length to be 21 or 22-nt. Given this general finding of a major determinant of on-target guide activity, we then focused on specificity of SaCas9 given these different spacer lengths, as 653 guide-target pairs is still too small of a dataset to solve the on-target prediction problem. We note, however, that the pairwise library methodology described here could be exploited for a much larger (e.g. 90,000 guide-target pairs) on-target efficiency screen to better explain variability within one spacer length.

Minor comments:

- Table 1 listed in the text actually appears to be Supplementary Table 1 (or Table 1 is missing.)

Thank you for this helpful correction. We apologize for the error and have corrected the text to state “Supplementary Table 1”.

Reviewers' comments:

Reviewer #1 (Remarks to the Author):

The authors have adequately addressed the concerns previously raised by this reviewer, returning with a substantially improved manuscript.

Reviewer #2 (Remarks to the Author):

In their revised manuscript the authors have added additional detail that has clarified important concerns that we had about the original manuscript – in particular the threshold that was used to define target sites that would be used in the calculation of the on-target vs off-target ratios. They have also added an additional timepoint for the editing analysis (14 days) that shows higher overall editing rates as well as consistency with the 3 day data for the impact of the guide length on on-target and off-target editing rates for single mismatches.

However, there are still some outstanding questions about the interpretation of their data:

Dichotomy of the highly active sites versus lower active sites for the estimation of off-target editing rates. We may not have phrased our initial question well in the original review – do highly active on-target sites have higher inherent off-target editing rates? The authors discuss some aspects of this in their rebuttal: “We see a weak but significant correlation ($R^2 = 0.16$, $p < .0001$) between the on-target activity and off-target activity of the guide-target pairs with single mismatches. The correlation is stronger ($R^2 = 0.35$) if we consider only PAM-distal mismatches, as expected. But, we see no correlation when we consider guide-target pairs with 2 mismatches or bulges. Overall, there is little correlation because the vast majority of the off-target sites are double-mismatch sites that are minimally edited by SaCas9 ”

This response does not appear consistent with their new 14 day dataset. The on-target activity of their guides as a function of spacer length increases from day 3 (Fig 1E) to day 14 (Sup Fig 2A) as would be expected, and there is a corresponding increase in the off: on-target editing ratios for the day 14 versus day 3 data (Sup Fig 2C). It is hard to imagine that this is entirely a saturation effect since the on-target editing rates in the 14 day sample are still moderate. Thus, it would appear from these data that there is some influence of the on-target editing rate on the observed off-target rate, which would not be unexpected. This is important, as the author's current interpretation of the data argues that for the 21 nt spacers, while they are the most active, they may be a poorer choice than other lengths because of the higher editing rates at simulated off-target sites. Thus the authors recommend that: “Most often, 22 nt spacers offer the optimal balance of efficiency and specificity, while 20 nt spacers improve specificity at the cost of lesser efficiency” However, if the reason that the 21 nt spacer has the highest apparent off-target activity is simply because this is guide length that is the most active in this assay, then if a similar amount of editing was present at a target site for the 20 and 22 nt length guides, they would have similar off-target rates.

The key thing that this reviewer is concerned about, is if the authors are promoting the use of less active guides to the community for their editing projects (basic science or therapeutic), it needs to be clear that the apparent improvement in specificity is not a result of difference in activity level for the guides in this assay, as all users will be trying to maximize the on-target editing rates for their applications. If the authors take the top 5% of guides in the 21 and 22 length groups and examine the off: on-target ratios, are they similar or different in specificity?

Minor note: on page 4 the authors focus on the day 3 results for their double mismatch data: “21 nt spacers were also more tolerant of double mismatches than any other spacer length ($p < 0.0001$, Dunn's multiple comparisons test, Supplementary Fig. 2C), in a position-dependent manner (Fig. 2C, E). 21 nt spacers had a mean 16% off: on target activity ratio while we observed a mean of

only 2% and 6% off: on-target activity across all double-mismatched sites targeted by 20 nt or 22 nt spacers, respectively (Fig. 2C, D), at Day 3." Given their new 14 day data (Sup Figure 2) it would be good to include a statement in the text about the increase in off-target rates for spacer lengths 22 -24 in this dataset, as the 22 nt spacer appears to have off-target rates that are quite similar to the 21 nt length guides.

Reviewers' comments:

Reviewer #1 (Remarks to the Author):

The authors have adequately addressed the concerns previously raised by this reviewer, returning with a substantially improved manuscript.

We thank the reviewer for their helpful feedback.

Reviewer #2 (Remarks to the Author):

In their revised manuscript the authors have added additional detail that has clarified important concerns that we had about the original manuscript – in particular the threshold that was used to define target sites that would be used in the calculation of the on-target vs off-target ratios. They have also added an additional timepoint for the editing analysis (14 days) that shows higher overall editing rates as well as consistency with the 3 day data for the impact of the guide length on on-target and off-target editing rates for single mismatches.

However, there are still some outstanding questions about the interpretation of their data:

Dichotomy of the highly active sites versus lower active sites for the estimation of off-target editing rates. We may not have phrased our initial question well in the original review – do highly active on-target sites have higher inherent off-target editing rates? The authors discuss some aspects of this in their rebuttal:” We see a weak but significant correlation ($R^2 = 0.16$, $p < .0001$) between the on-target activity and off-target activity of the guide-target pairs with single mismatches. The correlation is stronger ($R^2 = 0.35$) if we consider only PAM-distal mismatches, as expected. But, we see no correlation when we consider guide-target pairs with 2 mismatches or bulges. Overall, there is little correlation because the vast majority of the off-target sites are double-mismatch sites that are minimally edited by SaCas9 ” This response does not appear consistent with their new 14 day dataset.

The on-target activity of their guides as a function of spacer length increases from day 3 (Fig 1E) to day 14 (Sup Fig 2A) as would be expected, and there is a corresponding increase in the off:on-target editing ratios for the day 14 versus day 3 data (Sup Fig 2C). It is hard to imagine that this is entirely a saturation effect since the on-target editing rates in the 14 day sample are still moderate. Thus, it would appear from these data that there is some influence of the on-target editing rate on the observed off-target rate, which would not be unexpected.

This is important, as the author’s current interpretation of the data argues that for the 21 nt spacers, while they are the most active, they may be a poorer choice than other lengths because of the higher editing rates at simulated off-target sites. Thus the authors recommend that: “Most often, 22 nt spacers offer the optimal balance of efficiency and specificity, while 20 nt spacers improve specificity at the cost of lesser efficiency” However, if the reason that the 21 nt spacer has the highest apparent off-target activity is simply because this is guide length that is the most active in this assay, then if a similar amount of editing was present at a target site for the 20 and 22 nt length guides, they would have similar off-target rates.

The key thing that this reviewer is concerned about, is if the authors are promoting the use of less active guides to the community for their editing projects (basic science or therapeutic), it needs to be clear that the apparent improvement in specificity is not a result of difference in activity level for the guides in this assay, as all users will be trying to maximize the on-target editing rates for their applications. If the

authors take the top 5% of guides in the 21 and 22 length groups and examine the off:on-target ratios, are they similar or different in specificity?

We thank the reviewer for their helpful feedback and excellent suggested analysis. We analyzed the off:on-target ratios across all single mismatches for the top 10 most-active guides in the 20, 21, and 22 nt spacer groups. We used 'top 10' instead of 'top 5%' as 5% of 73 guide groups would leave only 3 guides per length. Within this subset, there was no significant difference in the on-target efficiency between the spacer lengths when we applied Dunn's multiple comparisons test. With a Mann-Whitney test between just length 20 and 21 nt spacer groups, the on-target efficiency of the 20 nt spacer was significantly lower on Day 3 ($p < .05$) but not on Day 14 ($p > .05$).

Next, we examined the specificity of these most-active guides and saw that the 20 nt spacer group remained significantly less tolerant of single mismatches, but the 21 and 22 nt spacer groups were not significantly different. (Only some of these guides were among those for which we tested double mismatches and bulges, so we did not repeat those analyses for the top 10 guides.)

The reviewer's question has led to an important clarification, and we are including a new figure (**Supplementary Fig. 3**) and text in the Results and Discussion (highlighted) to describe it. In short, we observe that highly-active 20 nt spacers are more rare, but they are still less tolerant of mismatches than similarly active 21 nt spacers. This parallels the higher specificity of 17-18 nt spacer SpCas9 'truncated guides'. We have revised the text so it is no longer suggesting that 22 nt spacers can achieve both improved specificity and equally high efficiency as the 21 nt spacers. Especially for therapeutic editing, it is worthwhile to screen for a highly active 20 nt spacer and then take advantage of their higher specificity.

As further validation, we note that we used a GFP knockout assay and observed similar on-target efficiency between 20 and 21 nt spacers, and again the 20 nt was much more specific (**Supplementary Fig. 6**).

Minor note: one page 4 the authors focus on the day 3 results for their double mismatch data: "21 nt spacers were also more tolerant of double mismatches than any other spacer length ($p < 0.0001$, Dunn's multiple comparisons test, Supplementary Fig. 2C), in a position-dependent manner (Fig. 2C, E). 21 nt spacers had a mean 16% off:on target activity ratio while we observed a mean of only 2% and 6% off:on-target activity across all double-mismatched sites targeted by 20 nt or 22 nt spacers, respectively (Fig. 2C, D), at Day 3." Given their new 14 day data (Sup Figure 2) it would be good to include a statement in the text about the increase in off-target rates for spacer lengths 22 -24 in this dataset, as the 22 nt spacer appears to have off-target rates that are quite similar to the 21 nt length guides.

We thank the reviewer for pointing out that we did not sufficiently interpret the Day 14 double mismatch data in the text and apologize for this omission. We agree that there is an increase in all lengths at Day 14, including 22-24, and now explicitly state this in the text. We also edited the paragraph about the double mismatches to focus on the difference between the 20 nt and the longer spacers. We think this is an improvement over the previous version which included comparisons between the 21 nt and 22 nt spacers. We thank the reviewer for their feedback that led to these improvements.

REVIEWERS' COMMENTS:

Reviewer #2 (Remarks to the Author):

The authors have fully addressed my concerns.